# Effects of TNFα receptor TNF-Rp55- or TNF-Rp75- deficiency on corneal neovascularization and lymphangiogenesis in the mouse

**Anna-Karina B. Maier** ⓘ *☉, **Nadine Reichhart**☉, **Johannes Gonnermann, Norbert Kociok, Aline I. Riechardt, Enken Gundlach, Olaf Strauß, Antonia M. Joussen**

Department of Ophthalmology, Charité–Universitätsmedizin Berlin, corporate member of Freie Universität Berlin, Humboldt- Universität zu Berlin and Berlin Institute of Health, Berlin, Germany

☉ These authors contributed equally to this work.
* anna-karina.maier@charite.de

**Data Availability Statement:** All relevant data are available on Zenodo (DOI: 10.5281/zenodo. 4049178).

## Abstract

Tumor necrosis factor (TNF)α is an inflammatory cytokine likely to be involved in the process of corneal inflammation and neovascularization. In the present study we evaluate the role of the two receptors, TNF-receptor (TNF-R)p55 and TNF-Rp75, in the mouse model of suture-induced corneal neovascularization and lymphangiogenesis. Corneal neovascularization and lymphangiogenesis were induced by three 11–0 intrastromal corneal sutures in wild-type (WT) C57BL/6J mice and TNF-Rp55-deficient (*TNF-Rp55d*) and TNF-Rp75-deficient (*TNF-Rp75d*) mice. The mRNA expression of VEGF-A, VEGF-C, Lyve-1 and TNFα and its receptors was quantified by qPCR. The area covered with blood- or lymphatic vessels, respectively, was analyzed by immunohistochemistry of corneal flatmounts. Expression and localization of TNFα and its receptors was assessed by immunohistochemistry of sagittal sections and Western Blot. Both receptors are expressed in the murine cornea and are not differentially regulated by the genetic alteration. Both *TNF-Rp55d* and *TNF-Rp75d* mice showed a decrease in vascularized area compared to wild-type mice 14 days after suture treatment. After 21 days there were no differences detectable between the groups. The number of VEGF-A-expressing macrophages did not differ when comparing WT to *TNF-Rp55d* and *TNF-Rp75d*. The mRNA expression of lymphangiogenic markers VEGF-C or LYVE-1 does not increase after suture in all 3 groups and lymphangiogenesis showed a delayed effect only for *TNF-Rp75d*. TNFα mRNA and protein expression increased after suture treatment but showed no difference between the three groups. In the suture-induced mouse model, TNFα and its ligands TNF-Rp55 and TNF-Rp75 do not play a significant role in the pathogenesis of neovascularisation and lymphangiogenesis.

## Introduction

Corneal neovascularization and lymphangiogenesis can be induced by various triggers including limbal insufficiency, inflammation, trauma or surgical manipulations [1,2]. This leads to

**Funding:** AKM: Financial support was provided for Anna-Karina B. Maier by the "Friedrich C. Luft" Clinical Scientist Pilot Program funded by Volkswagen Foundation and Charité Foundation and by the "Lydia Rabinowitsch-Stipendium" funded by Charité Universitätsmedizin Berlin. The funders had no role in study design, data collection and analysis, decision to publish, or preparation of the manuscript.

**Competing interests:** The authors have declared that no competing interests exist.

reduced transparency of the cornea, loss of visual acuity and higher risk for graft rejection after corneal transplantation [1,3,4]. Cytokines and growth factors orchestrate the cells involved in the development of new blood and lymph vessels. Major regulators of both inflammation-driven neovascularization and lymphangiogenesis are growth factors of the vascular endothelial growth factor (VEGF) family (VEGF A, C and D) [1,5].

Tumour necrosis factor α (TNFα), a pro-inflammatory cytokine, is known as a key regulator in inflammatory processes and is produced by various cell types, including neutrophils, macrophages, lymphocytes and endothelial cells [6,7]. In the cornea, TNFα is involved in inflammatory and neovascular processes during corneal wound healing [2,8–10]. Its levels have been found increased in the corneal epithelium, stroma and endothelium in murine and in human corneas during ocular surface inflammation [11]. The role of TNFα in corneal angiogenesis, however, is controversially discussed: Saika et al. showed that alkali burn-induced corneal neovascularization was more severe in $TNF\alpha^{-/-}$ mice than in wild-type mice [10]. Furthermore, $TNF\alpha^{-/-}$ mice showed more prominent central stromal neovascularization, accompanied by increased expression of transforming growth factor (TGF)-β1 and VEGF-A compared with wild-type mice [2]. In contrast, other reports discussed, that TNFα plays a role in inducing corneal neovascularization. Cade et al. and Fujita et al. showed that TNFα inhibition reduced corneal neovascularization in different animal models [2,8]. Furthermore, it is suggested that TNFα induces VEGF-A expression in macrophages recruited to the injured cornea [7].

Besides neovascularization, inflammatory processes in the cornea are accompanied by lymphangiogenesis [12]. Many studies demonstrated that macrophages secreting VEGF-C/VEGF-D are involved in the development of lymph vessels [13,14]. Zhang et al. showed that TNFα can stimulate lymphangiogenesis by inducing VEGF-C production in macrophages via NF-κB [15]. In another publication by Ji et al. TNFα stimulated the expression of VEGF-C/VEGFR-3 on corneal dendritic cells and macrophages [16,17]. Moreover, the inhibition of TNFα reduced significantly corneal neovascularization and lymphangiogenesis in the mouse model of ocular surface scarring [18].

TNFα can bind to two different cell surface receptors, TNF-receptor (TNF-R) p55 and TNF-Rp75. In general, TNF-Rp55 activation creates a pro-inflammatory environment, whereas TNF-Rp75 also shows anti-inflammatory effects [19,20].

The exact role of the two receptors in the eye, is still under investigation. Concerning the cornea, Lu et al. demonstrated in the alkali-burn model that *TNF-Rp55d* exhibited impaired corneal neovascularization through reduced expression of VEGF-A and iNOS by infiltrating macrophages [7].

In the present study we assessed the role of TNF-Rp55 and TNF-Rp75 in the model of suture-induced inflammatory corneal neovascularization and lymphangiogenesis using TNF-Rp55 deficient mice (*TNF-Rp55d)* and TNF-Rp75 deficient mice (*TNF-Rp75d*). We compared blood- and lymph vessel growth as well as TNFα expression.

## Material and methods

### Human tissue

Human cornea tissue was obtained from two patients after perforating keratoplasty. In both cases the perforating keratoplasty was performed because of a graft failure after keratoplasty due to a traumatic scar. Both patients showed a significant neovascularization of the graft. Written informed consent was obtained from both patients and approved by the Charité-Universitätsmedizin Berlin. The study followed the principles of the Declaration of Helsinki.

## Animals

All animal experiments adhered to the ARVO Statement for the Use of Animals in Ophthalmic and Vision Research and were approved by responsible University Animal Care and Use Committees, LAGESO (G 0326/12). Mice with deficient function of TNFR1 (*TNF-Rp55d*) (B6.129-Tnfrsf1a*tm1Mak*/J, Jackson lab stock 002818) or TNFR2 (*TNF-Rp75d*) (B6.129S2-Tnfrsf1b*tm1Mwm*/J, Jackson lab stock 002620) were used in the study. The mice harbor a null mutation of the respective TNFR leading to an altered gene product that lacks the molecular function of the wild-type gene. The proteins themselves are expressed.

The TNF-Rp55 mutation was created by disruption of the coding sequence of the TNF-Rp55 gene by insertion of a neo gene [21]. This resulted in the transcription and transduction of a non-functional TNF-Rp55 protein. A similar strategy was pursued by Erickson et al creating the *TNF-Rp75d* mice. A neomycin-resistance gene under control of the Pgk promoter20 was inserted in the second exon, which contains the signal peptide region of TNF-R2, resulting in the transcription and transduction of a non-functional TNF-Rp75 protein [22].

The genotype was determined by PCR analysis of genomic DNA prepared from tail or ear samples according to Rothe et al. [23]. Age matched C57BL/6J mice (Janvier, France) served as controls. A total number of 47 *TNF-Rp55d* mice, and 47 *TNF-Rp75d mice* and 48 wild-type mice were used for this study.

Animals were fed regular laboratory chow and water ad libitum. A 12-hour day–night cycle was maintained. After suture placement, the mice were regularly examined by animal keepers. All efforts were made to minimize discomfort of the animals.

## Mouse model of suture-induced, inflammatory corneal neovascularization

Corneal neovascularization was induced according to a standard protocol [5,24,25]. The contra-lateral eye served as control.

In brief, mice were deeply anesthetized by injection of ketamine/xylazine. The central cornea was marked by a 2 mm trephine, gently placed at the central cornea. Three 11–0 sutures were placed intrastromally with 2 incursions extending over 120 degrees of corneal circumferences each. The outer point of suture placement was chosen as halfway between the limbus and the line outlined by the 2 mm trephine, the inner suture point was equidistant from the 2 mm trephine line to obtain standardized angiogenic response. 3, 8, 14 or 21 days after suture, mice were sacrificed by injection of ketamine/xylazine and subsequent cervical dislocation, and eyes were processed for further experiments.

## Immunohistochemistry on sagittal sections and cornea whole-mounts

**Whole-mounts.** Eyes were enucleated and fixed in acetone for 8 minutes. The sclera was dissected with a circumferential incision parallel to the limbus, followed by removal of the lens and iris. Four radial cuts were made to allow flattening.

The whole-mount immunohistology protocol from [1] was slightly modified as follows: Corneas were blocked in 5% BSA in PBS with 10% goat serum and 0.1% Triton X-100 overnight. Then the corneas were incubated in primary antibody (Lyve-1 (1:500, Cat# DP3513P), Acris Antibodies, Germany) in 5% BSA in PBS for 3 days at 4˚C. A Cy3-conjugated species appropriate secondary antibody (Jackson ImmunoResearch Laboratories, USA) was used applied overnight at 4˚C in the same buffer. After PBS wash, corneas were incubated in FITC-conjugated CD31 (1:200) (Cat # 558738 BD Pharmingen, USA) overnight at 4˚C and mounted flat on slides.

Images of the flatmounts were captured with an Axio Imager 2 (Zeiss, Germany). Pictures were digitalized using ZEN software (Zeiss). The areas covered with blood and lymph vessels were detected with an algorithm and calculated using the software ImageJ (NIH, USA); a ratio

of vessel covered area to total area was calculated; prior to analysis, gray-value images of whole mount pictures were modified by several filters, and vessels were detected by threshold setting, including the bright vessels and excluding the dark background. Analysis was carried out in a masked fashion by two independent observers. The mean vascularized area of the control was set to 100%, and the vascularized areas of the other samples are given in relation to this value.

**Sagittal sections.** Enucleated mouse eyes and human corneas were fixed in 4% PFA overnight and embedded in paraffin. After sectioning, the samples were rehydrated and subjected to heat mediated antigen retrieval. After blocking the sections with 5% BSA in TBS for 1h at RT they were incubated in primary antibodies overnight at 4˚C: anti-CD68 (1:100; Cat # M0876; DAKO, Denmark); anti-VEGF-A (1:200; Cat # ab46154; abcam, UK) anti-F4/80 (1:200; Cat # ab16911; abcam); anti-vWF (1:100; Cat # A0082; DAKO); anti-TNF-Rp55 (1:100; Cat # AP06465PU-N; Acris, USA); anti-TNF-Rp75 (1:100; Cat# AP15825PU-M; Acris, USA).

After three washing steps, species appropriate fluorescent secondary antibodies were applied for 1h at RT. Nuclei were stained with DAPI. Subsequently sections were mounted on glass slides and subjected to an Axio Imager 2 (Zeiss, Germany). Pictures were digitalized using ZEN software (Zeiss).

## RNA Isolation and RT-PCR

Freshly isolated corneas were stored in RNAlater. The whole corneas were homogenized in lysis buffer (RNeasy Kit, Qiagen, Germany). RNA isolation and cDNA synthesis were performed according to the recommendations of the manufacturer (Qiagen).

The mRNA levels for VEGF-A, VEGF-C, Lyve-1, TNFα, TNF-Rp55, TNF-Rp75 and the calibration gene GAPDH in the mouse corneas after sutures were analyzed by Real Time RT-PCR using the Rotor-Gene SYBR Green PCR Kit (Qiagen) on a Rotor-Gene Q (Qiagen). Primer sequences are listed in Table 1. mRNA expression of the genes of interest together with the calibration gene were analyzed simultaneously in triplet reactions. The analysis was repeated two to four times. To confirm amplification specificity the PCR products from each primer pair was subjected to a melting curve analysis (for details see data repository). Genomic DNA contamination was excluded by choosing primers hybridizing to different exons or spanning exon borders. Moreover, control amplification reactions that were performed with non-transcribed RNA as templates gave only background fluorescence. Quantification of the calibrated target genes was done with the Rotor-Gene Q software 2.2.3 (Qiagen) applying the comparative CT (threshold cycle, CT) as described (Morrison et al., 1998).

**Western blot analysis of TNFα and its receptors.** Mouse corneas were homogenized in lysis buffer containing 100mM Tris-HCl and SDS 1%. Lysates were separated by 12% SDS-PAGE and transferred to a PVDF membrane. After blocking in 5% non-fat dried milk powder in TBS/Tween 0.05% 1 h at room temperature, the membranes were incubated overnight at 4˚C with primary antibody anti-TNFα (1:500; Cat # ABIN677318; antibodies-online

**Table 1. Primer sequences.**

| Gene | Amplicon | Forward Sequence (5' -> 3') | Reverse Sequence (5' -> 3') |
|---|---|---|---|
| VEGF-A | 201 | CAGCTATTGCCGTCCGATTGAGA | TGCTGGCTTTGGTGAGGTTTGAT |
| VEGF-C | 120 | AGCTGAGGTTTTTCTCTTGTGATTTAA | TGATCACAGTGAGCTTTACCAATTG |
| Lyve-1 | 174 | AGGAGCCCTCTCCTTACTGC | ACCTGGAAGCCTGTCTCTGA |
| TNFα | 94 | GCCTCCCTCTCATCAGTTCTAT | TTTGCTACGACGTGGGCTA |
| TNF-Rp55 | 154 | TGCGGTGCTGTTGCCCCTGGTTAT | CTTTCCAGCCTTCTCCTCTTTGA |
| TNF-Rp75 | 228 | CCCTACAAACCGGAACCTGG | CACCTGGTCAGTGGTACAGG |
| GAPDH | 190 | TTGTGCAGTGCCAGCCTC | TTGCCGTGAGTGGAGTCATAC |

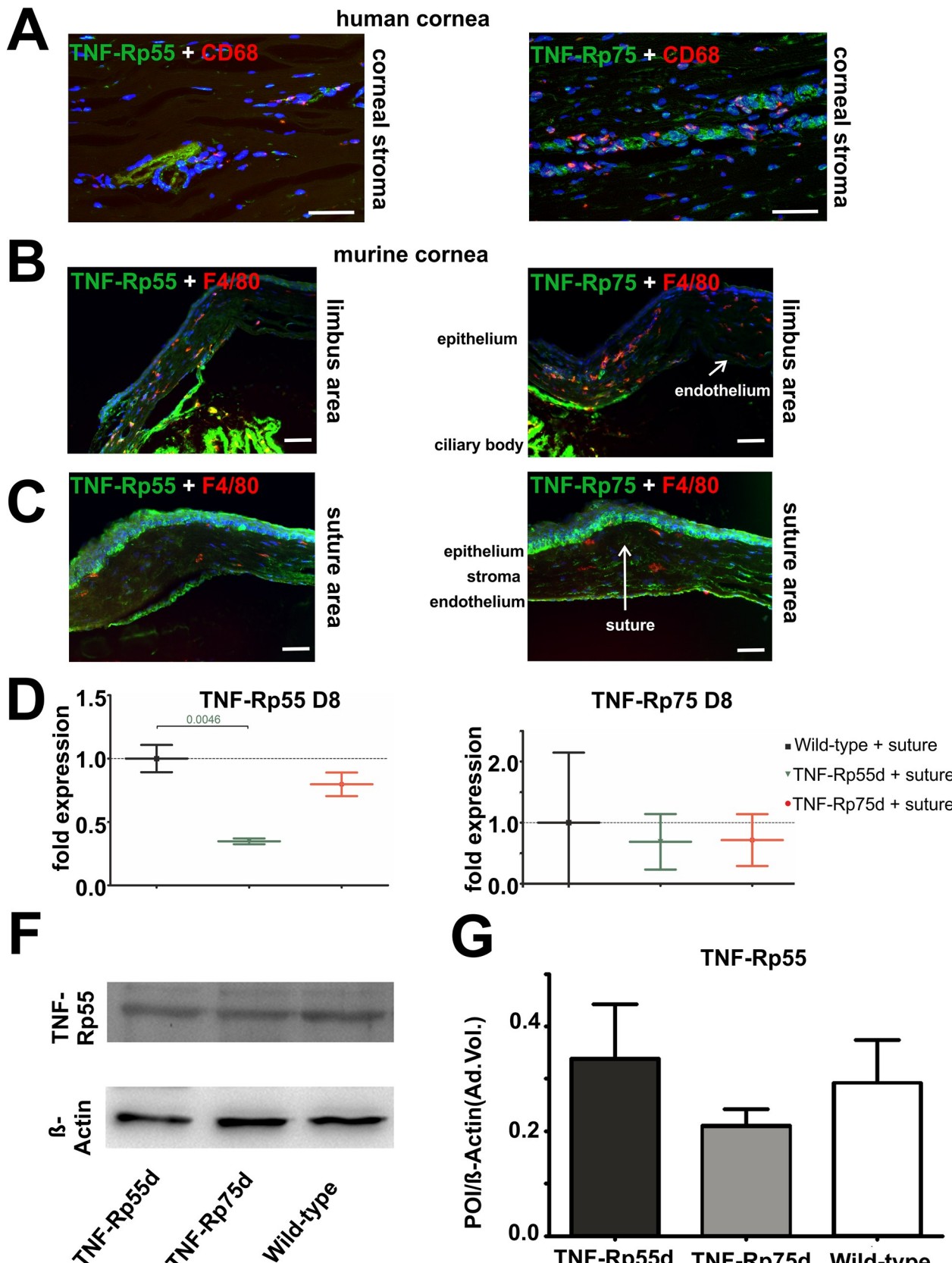

**Fig 1. TNF-Rp55 and TNF-Rp75 expression in human and in murine cornea. A:** Sagittal sections of vascularized human cornea stained with antibodies against TNF-Rp55 and TNF-Rp75 (green), and CD68 (red). Nuclei are stained with DAPI. Images display the stroma of human cornea.

**B, C:** Sagittal sections of wild-type murine cornea 14 days after suture intervention stained with antibodies against TNF-Rp55 and TNF-Rp75 (green), and F4/80 (red). Nuclei are stained with DAPI. Scale bar represents 50 μm. **D, E:** Differential mRNA expression of TNF-Rp55 (**D**) and TNF-Rp75 (**E**) in wild-type and *TNF-Rp55d* and *TNF-Rp75d* mice. Data are normalized to wild-type expression values (= 1); n = 6–9. **F, G:** Detection of TNF-Rp55 in representative western blot of corneal lysates from wild-type, *TNF-Rp55d* mice and *TNF-Rp75d* mice without or with suture intervention at day 14 (**B**). β-Actin served as loading control. Bar charts illustrates densitometric analysis of relative protein expression of soluble TNF-Rp55 /β-Actin (**C**); n = 3. The uncropped blots are shown in the data repository.

Inc., USA), anti-TNF-Rp55 (1:100; Cat # AP06465PU-N; Acris) and anti-β-Actin (1:10000, Cat # ab8224, abcam), diluted in TBS/Tween 0.05%. Membranes were washed three times for 10 minutes in TBS/Tween 0.05% and then incubated with the appropriate anti-rabbit or anti-mouse IgG HRP-labelled antibody (1:5000) (GE Healthcare UK Limited, Buckinghamshire, UK) for 1 h at room temperature. Proteins were visualized via chemiluminescence reaction (Bio-Rad Laboratories, Hercules, CA, USA). Blots were digitalized using a ChemiDoc YRS Imager with the software QuantityOne (Bio-Rad, Laboratories,Germany) and densitometry was performed with ImageJ (NIH, USA).

### Statistical analysis

All results are expressed as the mean ± standard error of the mean (SEM). After testing for normal distribution of the data (Kolmogorov-Smirnov-Test), the data were compared by unpaired or paired T-test in case of a normal distribution. Otherwise a non-parametric test (Whitney-Mann-U or Wilcoxon) was used. Differences were considered statistically significant when p values were less than 0.05.

## Results

### Expression of TNF-Rp55 and TNF-Rp75 in the cornea

Both TNFα receptors (TNF-Rp55 and TNFRp75) are expressed in the stroma of the human cornea. We also detected macrophages by CD68 staining, the receptors and macrophages, however, do not appear to co-localize (Fig 1A, S1 Fig).

In the mouse model of suture induced neovascularisation, both receptors are expressed in the endothelium, in the epithelium, and in the corneal stroma both in the central area of the suture and at the limbus area of the cornea. Comparable to the human tissue, no co-localization was found between the receptors and the F4/80 positive macrophages (Fig 1B and 1C). mRNA expression of both receptors TNF-Rp55 and TNF-Rp75 can be detected in the wild-type as well as in the TNF-Rp55d and TNF-Rp75d mice. The mRNA expression of TNF-Rp55 in the *TNF-Rp55d* mice, however, is significantly reduced (Fig 1D and 1E). There is no evidence for differential regulation of TNF-Rp75 in *TNF-Rp55d* animals or vice versa, neither on mRNA nor on protein level (Fig 1D–1G, S1 Table).

### Corneal neovascularization in *TNF-Rp55d* mice and *TNF-Rp75d* mice in the suture model

To investigate the effect of *TNF-Rp55d* mice and *TNF-Rp75d* mice on corneal neovascularization qPCR of VEGF-A, the major angiogenic factor, was performed. At day 3 after suture, wild-type mice, *TNF-Rp55d* mice and *TNF-Rp75d* mice showed significantly higher mRNA expression of VEGF-A compared to untreated littermates (level = 1). At day 8 after suture placement only *TNF-Rp75d* mice showed upregulation in VEGF-A mRNA expression. At day 14 there was no change in VEGF-A expression detectable anymore in wild-type and both *TNF-Rpd* mice after suture placement (Fig 2A, S1 Table).

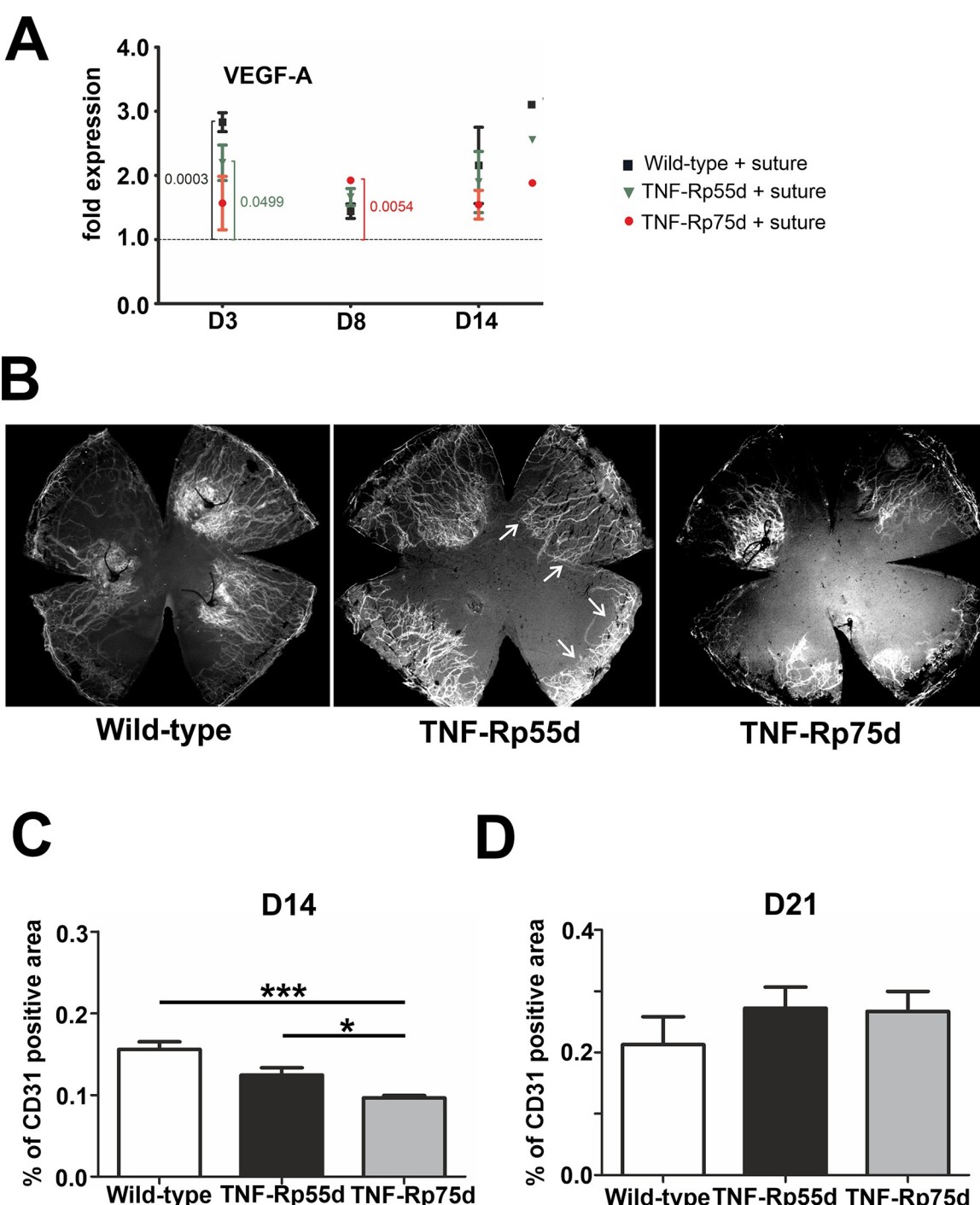

**Fig 2. Corneal neovascularization in *TNF-Rp55d* mice and *TNF-Rp75d* mice in the suture model. A:** Differential mRNA expression of VEGF-A in wild-type and *TNF-Rp55d mice* and *TNF-Rp75d* mice 3 days, 8 days, and 14 days after suture. Data are normalized to VEGF-A expression values of the respective genotype without suture (= 1); n = 2–9. **B:** Exemplary corneal whole-mounts of wild-type and *TNF-Rp55d* mice and *TNF-Rp75d* mice 14 days after suture placement. Blood vessels are stained with an antibody against CD31. Arrows depict CD31 covered area. **C, D:** Quantification of vascularized area compared to total corneal area in wild-type, *TNF-Rp55d* mice and *TNF-Rp75d* mice 14 days (C, n = 7–8) and 21 days (D, n = 4) after suture placement.

To localize neovascularization in the cornea after suture placement, blood vessel staining of corneal flatmounts was performed and the percentage of blood covered cornea was calculated 14 days and 21 days after suture placement. CD31 staining was applied to detect blood vessel covered area. The percentage of vascularized area in both *TNF-Rp55d* and *TNF-Rp75d* was significantly smaller compared to wild-type 14 days after suture placement (n = 7–8; p<0.001). Comparing *TNF-Rp55d* mice and *TNF-Rp75d* mice, the area covered with blood vessels was significantly smaller in *TNF-Rp75d* mice (n = 7–8; p = 0.029) (Fig 2B and 2C). At day 21, no significant differences between the mutant and wild-type animals in terms of CD31 coverage were detectable anymore (Fig 2D). There are no differences detectable in the number of CD68/VEGF-A positive macrophages in the limbus or the scar area among the three groups after suture placement (S2 Fig).

## Corneal lymphangiogenesis in *TNF-Rp55d* mice and *TNF-Rp75d* mice in the suture model

VEGF-C, a marker for lymphangiogenesis showed significant downregulation in the wild-type at day 14, whereas in *TNF-Rp55d* mice, VEGF-C was significantly downregulated at day 3. *TNF-Rp75d* mice did not show any significant changes in VEGF-C mRNA expression at the 3 distinct time points with or without suture placement (Fig 3A, S1 Table). In general, LYVE-1 mRNA expression did not change significantly in all the groups in the early phase (day3/8). Only at D14 the *TNF-Rp75d* mice showed significant upregulation of LYVE-1 after comparing untreated animals with sutured ones (Fig 3B, S1 Table). The area covered by lymphatic vessels (identified by LYVE-1 staining) was smaller in both genotypes, albeit only statistically significant for *TNF-Rp75d* mice compared to the wild-type after 14 days (n = 7–8; p = 0.014) (Fig 3D). Analogous to the blood vessel coverage, we also detected a significantly smaller percentage of LYVE-1 covered in *TNF-Rp75d* mice compared to *TNF-Rp55d* mice (Fig 3C and 3D).

## Influence of suture placement on TNF-α expression in wild-type, *TNF-Rp75d* and *TNF-Rp55d*

In wild-type animals suture placement led to a downregulation of TNFα mRNA expression at day 8 and 14 compared to control animals (Fig 4A). In *TNF-Rp55d* animals, this intervention caused a downregulation of TNFα expression at day 3 and day 14, but an upregulation at day 8. *TNF-Rp75d* mice showed an overall downregulation of TNFα mRNA expression upon suture placement until day 8 (Fig 4A, S1 Table).

Western Blot analysis at day 14 showed no significant changes in protein expression by suture placement in any of the genotypes (Fig 4B and 4C; S2 Table, S3 Fig).

## Discussion

Tumor necrosis factor (TNF)α and its two receptors, TNF-Rp55 and TNF-Rp75, are suspected to be involved in the process of inflammation and neovascularization in the cornea. Both receptors, TNF-Rp75 and TNF-Rp55 are expressed in the healthy cornea in humans and mice. In the present mouse study, suture intervention led to an increase in VEGF-A expression, whereas VEGF-C expression remained unchanged. *TNF-Rp75d* mice demonstrated more than *TNF-Rp55d* mice a subtle reduction of corneal neovascularization and lymphangiogenesis in the suture-induced inflammatory mouse model compared to wild-type mice only in the early phase after suture. After 21 days, there is no difference in the extent of area covered by blood vessels detectable anymore. There are no differences in the number of VEGF-A expressing macrophages when comparing *TNF-Rp55d* mice or *TNF-Rp75d* mice with wild-type controls.

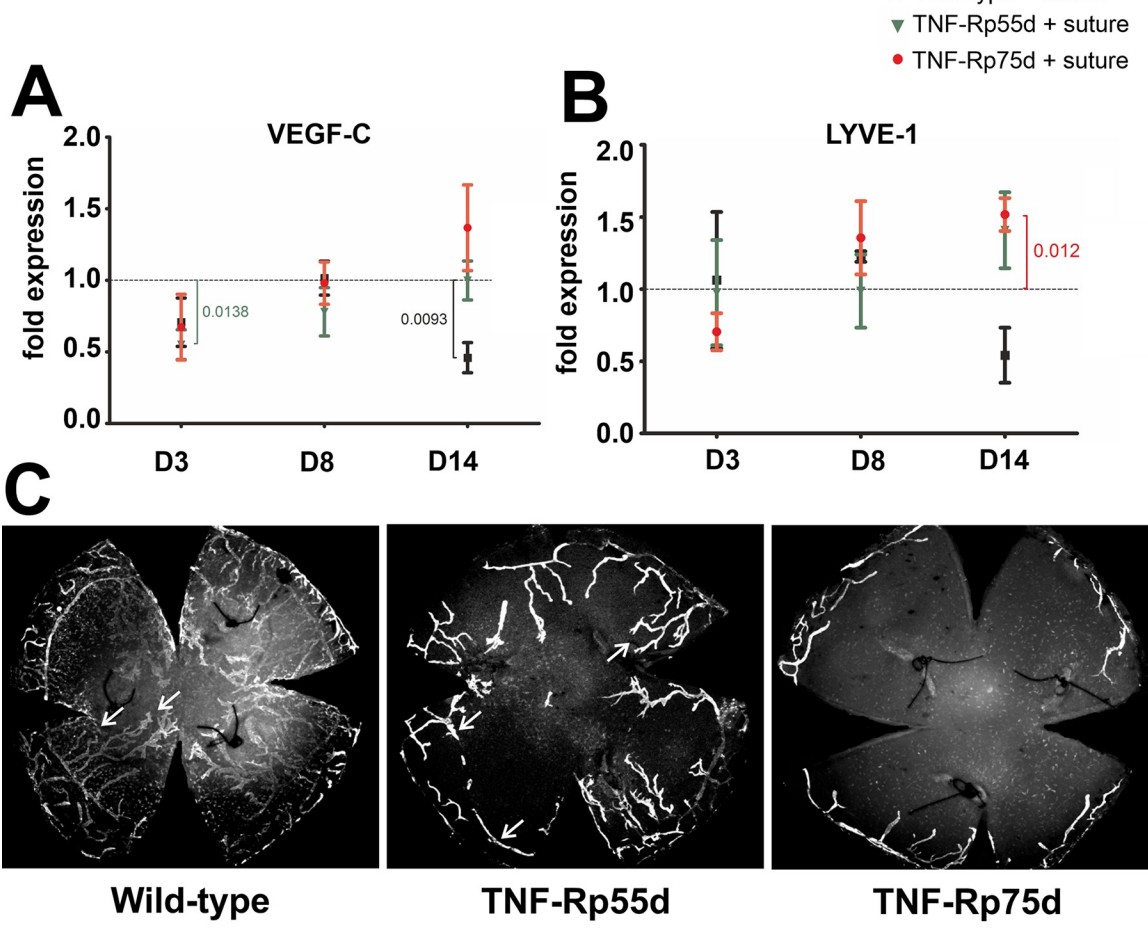

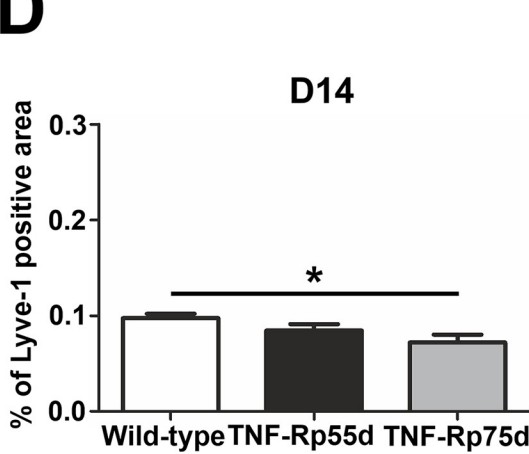

**Fig 3. Corneal lymphangiogenesis in *TNF-Rp55d* mice and *TNF-Rp75d* mice in the suture model. A, B:** Differential mRNA expression of VEGF-C (A) or LYVE-1 (B) in wild-type and *TNF-Rp55d* mice and *TNF-Rp75d* mice 3 days, 8 days, and 14 days after suture. Data are normalized to VEGF-A expression values of the respective genotype without suture (= 1); n = 2–3. **C:** Corneal flatmounts of wild-type and *TNF-Rp55d* mice and *TNF-Rp75d* mice 14 days after suture placement. Lymphatic vessels are visualized by Lyve-1-antibody. **D:** Quantification of percentage of LYVE-1 positive area compared to total corneal area in wild-type and *TNF-Rp55d* mice, and *TNF-Rp75d* mice 14 days after suture placement. n = 7–8.

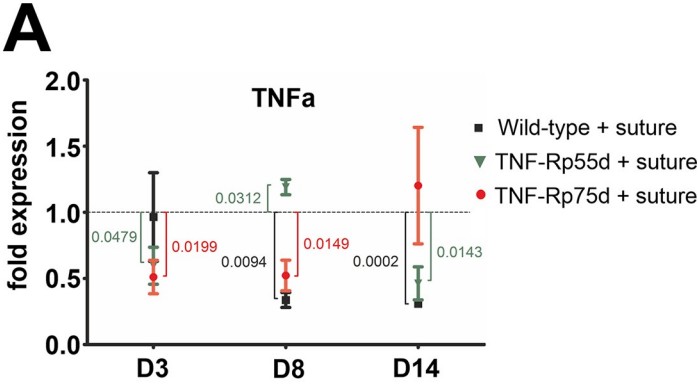

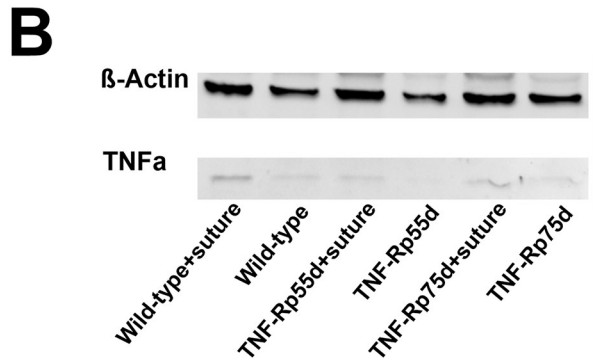

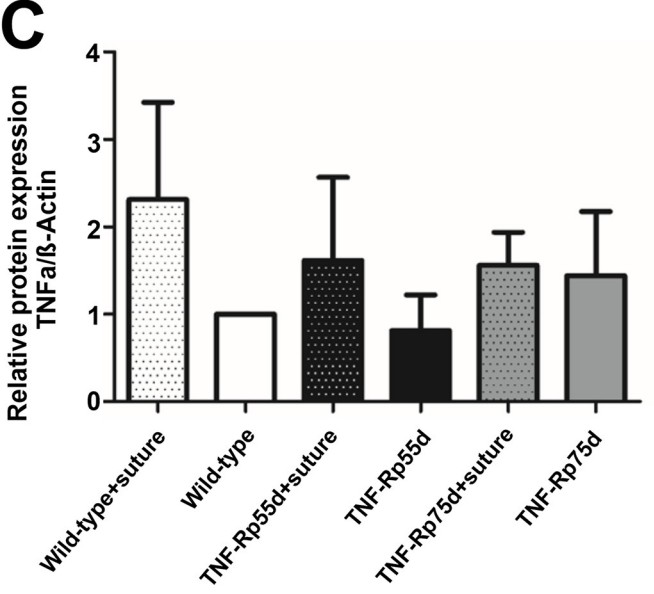

**Fig 4. Influence of suture placement on TNFα expression. A:** Differential mRNA expression TNF-α mRNA expression in wild-type and *TNF-Rp55d* mice and *TNF-Rp75d* mice 3 days, 8 days, and 14 days after suture. Data are normalized to VEGF-A expression values of the respective genotype without suture (= 1). n = 2–3. **B, C:** Detection of soluble TNF-α in representative western blot of corneal lysates from wild-type, *TNF-Rp55d* mice, and *TNF-Rp75d* mice without or with suture intervention at day 14 (**B**). β-Actin served as loading control. Bar charts illustrates densitometric analysis of relative protein expression of soluble TNF-α/β-Actin (**C**). n = 4 Uncropped blots are shown in S3 Fig. and in the data repository.

TNFα mRNA expression clearly increased after suture intervention but showed no differences among the different genotypes. Furthermore, there is no evidence that differential regulation of TNF-R expression accounts for the lack of effect in the mutant mice.

Taken together TNFα and thus its receptors TNF-Rp55 and TNF-Rp75 do not seem to play a direct nor crucial role in the pathophysiology in the mouse model of suture induced lymphangiogenesis and neovascularisation in the cornea.

Using another experimental paradigm, the mouse model of oxygen-induced retinopathy, our group detected only minor changes in the *TNF-Rpd* mice compared to wild-type controls. *TNF-Rp55d* and *TNF-Rp75d* mice presented a similar retinal development and vascularization under normoxic conditions without alterations in vascularization. Treatment with oxygen led to subtle reduction of vascularization in *TNF-Rp55d* mice on P17 and P20 [26].

We saw upregulation of VEGF-A mRNA expression in WT and *TNF-Rp55d* after suture intervention and subtle differences in the extent of blood vessel covered area at day 14 after suture in both *TNF-Rp75d* mice and *TNF-Rp55d* mice compared to wild-type. Using the alkali-burn model of corneal neovascularization, Lu et al showed that *TNF-Rp55d* mice presented reduced corneal neovascularization after 2 and 4 weeks compared to wild-type [7].

Additionally, Ferrari et al. showed that the inhibition of TNFα reduced significantly not only the corneal neovascularization, but also the lymphangiogenesis in the mouse model of ocular surface scarring [18]. Moreover, Zhang et al. demonstrated that TNFα is able to induce macrophages to produce VEGF-C, a lymphangiogenic factor, through NF-κB, further stimulating lymphangiogenesis [15]. Li et al. showed that TNFα stimulated the expression of VEGF-C/VEGFR-3 on corneal dendritic cells and macrophages [16,17]. In our study, the mRNA levels of VEGF-C and Lyve-1, the main cell surface receptor for hyaluronan, which mediated the proliferation, migration tube formation and signal transduction of lymphatic endothelial cells [27], did not show any significant changes before and after suture placement except a reduction of VEGF-C mRNA expression at day 3 after laser in the *TNF-Rp55d* mice. Regarding protein expression, we detected a smaller extent of lymph vessel covered area (LYVE-1 positive) in TNF-Rp75d mice compared to TNF-Rp55d and wild-type mice at day 14.

In the alkali-burn model TNFα mRNA expression increased in wild-type and *TNF-Rp55d⁻* mice to similar extents, which is comparable to our findings in the suture model. However, the effect was weaker in our approach, and on the protein level, we could not observe any changes upon suture intervention [7].

Altogether our findings are very subtle and mild compared to the previous results from other groups in terms of neovascularisation, lymphangiogenesis and also infiltration of VEGF-A expressing macrophages. Lu et al. suggested that the reduced neovascularization of *TNF-Rp55d* mice is mediated by reduced expression of VEGF and iNOS by infiltrating macrophages [7]. In contrast, we did not find differences in the number of VEGF-A expressing macrophages among the three groups. We even did not find any substantial hint that TNFα or its receptors are crucial for the induction of suture induced (lymph-) angiogenesis. In our approach, though significant on the mRNA level, the protein expression of TNFα in animals after suture was not significantly higher than in animals that did not undergo suturing.

In contrast to the alkali burn model or the surface scratch model, the suture intervention presents a relatively weak trigger to mimic corneal neovascularisation and angiogenesis upon tissue damage. The other models create a severe damage that induces massive inflammation and wound healing responses. So, it is possible to detect effects of their respective intervention on the ability of macrophages to secrete VEGF-A or VEGF-C by upregulation of TNFα, fostering the (lymph-) angiogenic response.

In opposite to the other models, we did not find substantial increase in TNF-a protein that induces secretion of angiogenic and inflammatory molecules by macrophages [7].

In summary, we conclude that in the mouse models of suture induced neovascularisation and lymphangiogenesis, TNFα, and its ligands TNFRp55 and TNF-Rp75 do not play a direct role in the pathogenesis.

However, the fact, that the suture-induced model represents a low-grade-inflammation model and shows a rather weak phenotype compared to the other models of corneal angiogenesis, also harbours several advantages: First, suture placement is more reproducible than alkali-burn or surface scratch. Thus several studies applying the two latter methods might lead to very different results depending on the intensity of the noxious intervention.

Second, it is more suitable to analyze lymphangiogenesis and associated secretion of pro-lymphangiogenic cytokines [28]. The induced neovascularisation and lymphangiogenesis mimic the clinical scenario of high-risk situation for a corneal graft rejection after keratoplasty and thus it has more translational impact than the models with severe damage and reduced stimulation of the lymphangiogenesis. Suturing only cause localized epithelial loss and inflammatory infiltration between the suture and the limbus, but chemical burns deplete the whole epithelial layer of the central cornea and cause strong cellular infiltration of the whole cornea [29].

## Supporting information

**S1 Fig. Isotype controls for the secondary antibodies applied in Fig 1, 1B and 1C and S2 Fig, respectively. A** (left side) depicts epithelium and stromal part of the cornea, while **A** (right side) depicts stroma and endothelium. Scale bars represent 100μm (**A**) or 50μm (**B**).
(TIF)

**S2 Fig. Immunohistochemical staining of corneal sections of sutured wild-type, *TNF-Rp55d*, and *TNF-Rp75d* mice in the suture and the limbus area using antibodies against VEGF-A (green) and CD68 (red).** Nuclei were stained with DAPI. Scale bar represents 50 μm.
(TIF)

**S3 Fig. Uncropped WB of TNFa (Fig 4). A, B:** Uncropped representative images of Western blots used for the densitometric analysis in Fig 4C. All uncropped Western blots are shown in the Data repository.
(TIF)

**S1 Table. Statistical analysis of the qPCR data.**
(DOCX)

**S2 Table. Statistics of the densitometric analysis.**
(DOCX)

**S1 Raw images.**
(PDF)

## Acknowledgments

The authors thank Gabriele Fels and Karin Oberländer for their valuable assistance.

## Author Contributions

**Conceptualization:** Anna-Karina B. Maier, Nadine Reichhart, Olaf Strauß, Antonia M. Joussen.

**Data curation:** Anna-Karina B. Maier, Nadine Reichhart, Norbert Kociok.

**Formal analysis:** Anna-Karina B. Maier, Nadine Reichhart, Johannes Gonnermann, Norbert Kociok.

**Investigation:** Anna-Karina B. Maier, Nadine Reichhart, Johannes Gonnermann, Norbert Kociok, Aline I. Riechardt, Enken Gundlach.

**Methodology:** Anna-Karina B. Maier, Nadine Reichhart, Johannes Gonnermann, Norbert Kociok, Aline I. Riechardt, Enken Gundlach.

**Supervision:** Olaf Strauß, Antonia M. Joussen.

**Visualization:** Anna-Karina B. Maier, Nadine Reichhart.

**Writing – original draft:** Anna-Karina B. Maier, Nadine Reichhart.

**Writing – review & editing:** Norbert Kociok, Olaf Strauß, Antonia M. Joussen.

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
