## [Decision Letter · Decision Letter 0]

17 Jul 2020

PONE-D-20-16932

Influence of the malfunction of the TNF-α receptors TNF-Rp55 or TNF-Rp75 on corneal neovascularization and lymphangiogenesis in the mouse

PLOS ONE

Dear Dr. Maier

Thank you for submitting your manuscript to PLOS ONE. After careful consideration, we feel that it has merit but does not fully meet PLOS ONE’s publication criteria as it currently stands. Therefore, we invite you to submit a revised version of the manuscript that addresses the points raised during the review process.

Please see the descriptions provided by the reviewers below. They all appear to have raised issues with experimental controls and data quality which should be addressed.

We look forward to receiving your revised manuscript.

Kind regards,

Christina L Addison, Ph.D.

Academic Editor

PLOS ONE

Journal Requirements:

2. To comply with PLOS ONE submissions requirements, in your Methods section, please provide additional information on the animal research and ensure you have included details on (a) methods of sacrifice, (b) methods of anesthesia and/or analgesia, and (c) efforts to alleviate suffering.

Reviewers' comments:

Reviewer's Responses to Questions

**Comments to the Author**

1. Is the manuscript technically sound, and do the data support the conclusions?

Reviewer #1: Partly

Reviewer #2: Yes

Reviewer #3: No

Reviewer #4: No

2. Has the statistical analysis been performed appropriately and rigorously? 

Reviewer #1: Yes

Reviewer #2: Yes

Reviewer #3: No

Reviewer #4: Yes

3. Have the authors made all data underlying the findings in their manuscript fully available?

Reviewer #1: Yes

Reviewer #2: Yes

Reviewer #3: Yes

Reviewer #4: Yes

4. Is the manuscript presented in an intelligible fashion and written in standard English?

Reviewer #1: Yes

Reviewer #2: Yes

Reviewer #3: Yes

Reviewer #4: Yes

5. Review Comments to the Author

Reviewer #1: In this study, Maier et al. investigate the potential role of TNFRs 1 and 2 in corneal neovascularization and lymphangiogenesis. The authors find a decrease in the vascularized area in TNFR-deficient animals as well as well as downregulation of TNF itself in their model. However, they conclude that TNF as well as its receptors do not play a significant role in the pathogenesis of neovascularization and lymphangiogenesis. The authors should consider to moderate this statement and their interpretation.

FIGURE 1

The data in figure 1 would be more convincing if a negative control was included. Since the authors have access to the TNFR-deficient mice, these samples should be included to demonstrate specificity of their antibodies.

What was the reason for using two different antibodies (CD68 and F4/80) in these samples?

It is not clear why the TNFR-deficient mice would show TNFR gene expression. The authors should validate their RT primers and repeat these experiments.

Also, the western blot in figure 1F is inconclusive. Why is their no difference between wild type and knock out? Immunoblots for TNFR1 should also be included.

FIGURE 2

CD31 positive/blood vessel area should be highlighted/indicated in figures 2B and 3C

FIGURE 4

The data in figure 4 is inconclusive and is hard to interpret as is. The quality of the immunoblot is rather low. The authors could also consider evaluating the function of TNF using an in vitro system. This would further strengthen interpretation and conclusion.

Minor comments:

The title is confusing and should be revised. The authors study the effects of deficiency rather than malfunction.

The overall quality of the figure-PDF file is low. High resolution images and unified labels (font + size) should be provided with the revised manuscript.

Reviewer #2: It is pleasing to read the report by Dr. Maier and colleagues in which the group tested the relative efficacy of two TNF receptor deletions on vascularization following corneal injury; the reporting of minimal or negative results is not done often enough and this finding herein is important. The mouse experiments and data are sound, with no problems in their presentation, interpretation, or subsequent discussion. There is some concern with the human data: it is very hard from the way the imaging studies were performed to claim co-localization or lack thereof from any particular cell type (the imaging problem exists with the mouse tissue as well). Since this paper is not presenting a tissue-specific deletion of either Rp55 or Rp75, it's relative localization is not critically important to the findings. Being purposefully vague might be the safest route, claiming that protein is expressed and 'does not appear' to localize with macrophages... the epithelium is usually 'hot' for fluorescence so that claim is questionable and S3 and S4 are not convincing (and lack proper labels and legends).

Reviewer #3: In this manuscript, the authors characterized corneal neovascularization and lymphangiogenesis in the TNF-Rp55 or TNF-Rp75 KO mice. They concluded that in the suture-induced mouse model, TNFα and its ligands TNF-Rp55 and TNF-Rp75

do not play a significant role in corneal neovascularization and lymphangiogenesis. However, the data showed that a reduction of neovascularization in these KO mice compared with WT at day 14 and reduced lymphangiogenesis in TNF-Rp75 KO mice. This suggests that TNF signaling has a role in these two processes.

Other comments:

1. In most experiments, how many mice were used is unclear.

2. The authors only explored the expression of 3 genes that are related to neovascularization and lymphangiogenesis. Since many genes involve in regulation of neovascularization and lymphangiogenesis, the authors can't make a conclusion that TNFα and its ligands TNF-Rp55 and TNF-Rp75 do not play a significant role.

3. TNF-a is an important cytokine in regulating corneal inflammation, neovascularization and lymphangiogenesis. In this study, a suture-induced corneal neovascularization model was used to identify the effect of TNF-a in corneal neovascularization and lymphangiogenesis. The model is so mild, many of the immunological cells and cytokines were not found significantly changed in this model. So that’s possible that you can’t observe much effects of TNF-a in this situation. Also, the suture-induced corneal neovascularization is not suitable to mimic the clinical scenario of corneal graft rejection. As we know, the corneal graft rejection is reasoned by exogenous antigen stimulation, which is not similar to the suture induced corneal neovascularization.

4. A group of non-injured mice need to be added in each experiment. The authors should identify the expression of TNF-a has been increased in your models. Otherwise, there is no point to investigate the roles of TNF-a and its receptors in this model.

Reviewer #4: The authors show that TNFα and its receptors are not required for neovascularization and lymphangiogenesis in corneal suture model in mice. The data presented in Fig1 E, F and G indicates that the authors are working with mice where the TNFR receptors are not knocked and thus cannot make any claim on the role of TNFR in angiogenesis and lymphangiogenesis. I have to recommend rejecting the paper in its current form.

The following issues need to be addressed.

1. The quality of tissue in immunofluorescence images Fig1A and S3 and resolution of image is very poor and cannot be interpreted. It is unclear what region of the eye is shown in Fig1A although from result section it appears to be cornea. There is no indication of where TNFα receptor are localizing in the cornea. Please indicate where the receptors are localizing.

2. Fig 1B suffers from the same issue as Fig1 A. The tissue section is of extremely poor quality. The limbus region is suspect. Please show a phase image so one might see the iridiocorneal angle. The labels in B and C are confusing is label for C on left makes no sense to the reader. Should it not be cornea? If B and C are from the suture experiment (says Mouse+Suture on the right of panels) where are the controls or sham eyes?

3. Why is TNFR2 75KDa not reduced in the TNFR2 knockout mouse in Fig 1D? Please reword Line 223-227 “The mRNA expression of TNFRp55 in the TNF-RP55d mice, however, is significantly reduced (Fig.1D, E) and the functionality of the receptors is disturbed. There is no evidence for differential regulation ofthe other receptor in the TNF-Rpd animals, neither on mRNA nor on protein level (Fig.1D, E,F, G, S1 Table 1) as isis very confusing.

4. Fig 1F and G leads to question the entire premise of the paper- it seems TNFR1 protein is present in mutant hence the authors are working with essentially wt mice?? What is the state of the TNFR2 mutant mice? Does it still express TNFR2 protein? Please validate mice before doing the suture experiments. Please show a western blot showing the protein levels of TNFR1 and TNFR2 in wt and respective mutant mice.

5. Also need to this experiment with the double knockout mice given TNFα knockout seems to exacerbate neovascularization (Fujita 2007).

6. The figures all seem to be of poor resolution making it really hard to zoom in on panels to see detail.

6. PLOS authors have the option to publish the peer review history of their article (what does this mean?). If published, this will include your full peer review and any attached files.

Reviewer #1: No

Reviewer #2: No

Reviewer #3: No

Reviewer #4: No

---

## [Author Response · Author response to Decision Letter 0]

26 Sep 2020

We thank the editorial board to give us the opportunity to improve our paper substantially according to the academic editor and reviewers’ comments. 

We answer the academic editor and reviewer’s concerns point by point as follows.

Answer: We revised the manuscript according to the PLOS ONE`s style requirements. 

2. To comply with PLOS ONE submissions requirements, in your Methods section, please provide additional information on the animal research and ensure you have included details on (a) methods of sacrifice, (b) methods of anesthesia and/or analgesia, and (c) efforts to alleviate suffering.

Answer: We included the details on (a) methods of sacrifice, (b) methods of anesthesia and/or analgesia, and (c) efforts to alleviate suffering.

 “After suture placement, the mice were regularly examined by animal keepers. All efforts were made to minimize discomfort of the animals.”

“In brief, mice were deeply anesthetized by injection of ketamine/xylazine.”

“3, 8, 14 or 21 days after suture, mice were sacrificed by injection of ketamine/xylazine and subsequent cervical dislocation, and eyes were processed for further experiments.” 

Answer: All data are available at the open access database Zenodo using the following link (https://doi.org/10.5281/zenodo.4049178)

Answer: We added Figure S3 Fig depicting 2 representative uncropped blots. Furthermore, all blot data are available at a public data repository (https://doi.org/10.5281/zenodo.4049178).

 S3 Fig

Answer: Data of melting curve analysis are available at a public data repository (https://doi.org/10.5281/zenodo.4049178).

Reviewer #1: 

Q1: In this study, Maier et al. investigate the potential role of TNFRs 1 and 2 in corneal neovascularization and lymphangiogenesis. The authors find a decrease in the vascularized area in TNFR-deficient animals as well as well as downregulation of TNF itself in their model. However, they conclude that TNF as well as its receptors do not play a significant role in the pathogenesis of neovascularization and lymphangiogenesis. The authors should consider moderating this statement and their interpretation.

Answer: We investigated the role of TNFα in the suture model at two levels: differential TNFα gene expression and aberration of TNF receptor function. TNFα expression remained unchanged after suture placement. Suture placement led to differential regulation of TNFα at the mRNA level, no significant changes were detectable in protein levels, though. No differences were detectable in TNFα comparing WT to mutants. Indeed, the reviewer is right that we found differences in vascularization (determined by CD31 coverage). The differences, however, were very subtle and only detectable at 1 time point (D14). The animal model should show that TNFα -signaling might play a role by increasing TNFα -sensitivity of participating cells at constant TNFα levels. Again, these data cannot provide evidence for a significant role of TNFα- signaling in the suture model. 

Fig1

Q2: The data in figure 1 would be more convincing if a negative control was included. Since the authors have access to the TNFR-deficient mice, these samples should be included to demonstrate specificity of their antibodies.

Answer: We agree with the reviewer that we did not show a control for immunohistochemistry by using specimen from TNFR deficient tissue. For that we would have to use TNFR KO mice, lacking the expression of the respective gene by transgenic knocked down. In our case, however, we use KI models leading to lack of function of the respective proteins. Here, TNF-Rp55 and TNF-Rp75 are produced but their functional activity was switched off by the transgene. For more details how the KI mutations were created see answer to Q4. An isotype control, depicting the background staining of the secondary antibody, is included to the supplement (S1 Fig). We added arrows and text to ease the understanding of the figure

Q3: What was the reason for using two different antibodies (CD68 and F4/80) in these samples?

Answer: We used CD68 predominantly for human samples to detect macrophages and F4/80 in murine samples since they are more convincing to identify all macrophages in mice [1]. Unfortunately, F4/80 is only detectable in rodents.

Q4: It is not clear why the TNFR-deficient mice would show TNFR gene expression. The authors should validate their RT primers and repeat these experiments.

Answer: This is a misunderstanding of our model. We have to apologize that the current description was not clear enough. Thus, we now clarify the nature of the mutant mice, their genetic background and the functional consequences more precise. As already stated above, the TNFR deficiency is a functional one. The mice harbor a null mutation of the respective TNFR leading to an altered gene product that lacks the molecular function of the wild-type gene. The non-functional proteins themselves are expressed, however. The TNF-Rp55 null mutations were created by disruption of the coding sequence of the TNF-Rp55 gene at base pair 535 (corresponding to the cDNA sequence) by insertion of a neo gene [2]. This resulted in the transcription and transduction of a non-functional TNF-Rp55 protein. Pfeffer et al validated the mutation by an experiment that showed the inability of spleen cells of the TNF-Rp55-/- mice to bind TNF-a

A similar strategy was pursued by Erickson et al creating the TNF-Rp75-/- mice [3]. In TNF-Rp75-/- mice challenged by LPS (that leads to shedding of the extracellular domain of TNF-Rp75), no soluble TNF-Rp75 could be detected in the serum. 

We added the following sentence in the methods part to explain precisely the KI Model (p.5 Line 116ff:

“The mice harbor a null mutation of the respective TNFR leading to an altered gene product that lacks the molecular function of the wild-type gene. The non-functional proteins themselves are expressed. The TNF-Rp55 mutation was created by disruption of the coding sequence of the TNF-Rp55 gene by insertion of a neo gene (21). This resulted in the transcription and transduction of a non-functional TNF-Rp55 protein. A similar strategy was pursued by Erickson et al creating the TNF-Rp75d mice. A neomycin-resistance gene under control of the Pgk promoter20 was inserted in the second exon, which contains the signal peptide region of TNF-R2, resulting in the transcription and transduction of a non-functional TNF-Rp75 protein (22)”

Q5: Also, the western blot in figure 1F is inconclusive. Why is there no difference between wild type and knock out? Immunoblots for TNFR1 should also be included.

We validated the role of the TNFα signaling at the level of the TNFα -receptors. Figure 1F is an immunoblot showing the protein expression of TNF-Rp55 in TNF-Rp55d, TNF-Rp75d mice and wild-type littermates. As shown in bar chart 1G, there is no difference in protein expression of TNF-Rp55 among the groups. Especially TNF-Rp55 is not upregulated in the TNFrp-55d mice. Thus, we have convincing evidence, for the lack of compensatory upregulation of the TNFα-receptors expression in the KI-animals.

Fig2

Q6: CD31 positive/blood vessel area should be highlighted/indicated in figures 2B and 3C

Answer: We indicated the CD31/Lyve-1 covered area by arrows 

Fig2 and Fig3

Q7: The data in figure 4 is inconclusive and is hard to interpret as is. The quality of the immunoblot is rather low. The authors could also consider evaluating the function of TNF using an in vitro system. This would further strengthen interpretation and conclusion.

Answer: We tried to evaluate the role of TNFα signaling in suture induced neovascularization and lymphangiogenesis at several levels. We analyzed gene expression and protein expression of TNFα and its receptors in wild-type and TNF-Rp55d or TNF-Rp75d mice respectively and could not find reliable evidence. Thus, taken our results from animal experiments, in vitro experiments would not add valuable data to the manuscript. More complex signaling cascades that involve many different cell types and structures, an environment that is impossible to mimic in vitro might help to explain the pathogenesis of suture induced alterations in the animals.

The protein concentration of TNF-α in the cornea is very low, also compared to other, mainly structural proteins that are expressed abundantly in the cornea. We provide an uncropped, unedited version of the original blot (see S3Fig). Unfortunately we are not able to substantially improve the quality of the signal. Even precipitation of TNFα did not improve the visibility of the TNF-α band.

Minor comments:

Q8: The title is confusing and should be revised. The authors study the effects of deficiency rather than malfunction.

Answer: We changed the title to “Effects of TNFα receptors TNF-Rp55 or TNF-Rp75- deficiency on corneal neovascularization and lymphangiogenesis in the mouse.

Q9: The overall quality of the figure-PDF file is low. High resolution images and unified labels (font + size) should be provided with the revised manuscript.

Answer: The reviewer is right; the images in the current version are of poor quality/resolution. We significantly improved the resolution of the images and unified all the labels. We think that the current version is more appropriate now.

Reviewer #2: It is pleasing to read the report by Dr. Maier and colleagues in which the group tested the relative efficacy of two TNF receptor deletions on vascularization following corneal injury; the reporting of minimal or negative results is not done often enough and this finding herein is important. The mouse experiments and data are sound, with no problems in their presentation, interpretation, or subsequent discussion. There is some concern with the human data: it is very hard from the way the imaging studies were performed to claim co-localization or lack thereof from any particular cell type (the imaging problem exists with the mouse tissue as well). Since this paper is not presenting a tissue-specific deletion of either Rp55 or Rp75, it's relative localization is not critically important to the findings. Being purposefully vague might be the safest route, claiming that protein is expressed and 'does not appear' to localize with macrophages... the epithelium is usually 'hot' for fluorescence so that claim is questionable and S3 and S4 are not convincing (and lack proper labels and legends).

Answer: We thank the reviewer for his interest in the manuscript and we are thankful for the comments to improve the manuscript. We know that the results of co-localization analysis on sagittal sections have to be handled carefully. We agree with the Reviewer to state our conclusion more vaguely. Thus, we rephrased the sentence on p.9 line 228f. It is less speculative now by only stating “We also detected macrophages by CD68 staining, the receptors and macrophages, however, do not appear to co-localize.” 

We renewed the negative controls by more convincing ones and added labels to S1 Fig (S3)/S2 Fig (S4) to ease the understanding at a glance.

Reviewer #3: In this manuscript, the authors characterized corneal neovascularization and lymphangiogenesis in the TNF-Rp55 or TNF-Rp75 KO mice. They concluded that in the suture-induced mouse model, TNFα and its ligands TNF-Rp55 and TNF-Rp75

do not play a significant role in corneal neovascularization and lymphangiogenesis. However, the data showed that a reduction of neovascularization in these KO mice compared with WT at day 14 and reduced lymphangiogenesis in TNF-Rp75 KO mice. This suggests that TNF signaling has a role in these two processes.

Answer: Indeed, the reviewer is right that we found differences in vascularization and lymphangiogenesis (determined by CD31 or LYVE-1 coverage). The differences, however, were very subtle and only detectable at 1 time point (D14). These subtle differences might be statistically significant but not biologically relevant: The exact values for CD31 coverage on day 14 are: WT 15.6% ± 2.6%, TNF-Rp55d 12.4% ± 2.6% and TNF-Rp75d 9.6% ± 0.8%; for LYVE-1 coverage WT 9.7% ± 1.3%, TNF-Rp55d 8.4% ± 1.9%, TNF-Rp75d 7.2% ± 2.2%. The animal model should show that TNFα -signaling might play a role by increasing TNFα -sensitivity of participating cells at constant TNFα levels. Again, these data cannot provide evidence for a significant role of TNFα- signaling in the suture model. 

Other comments:

Q1: In most experiments, how many mice were used is unclear.

Answer: we provide numbers for all individual figures now in the figure legend and the supplement (S1 Table/S2 Table for WB and PCR) as well as an overview of the total amount of animals in the methods part. We added this sentence (p5. Lines 127-128): “A total number of 47 TNF-Rp55d mice, and 47 TNF-Rp75d mice and 48 wild-type mice were used for this study.”

Q2: The authors only explored the expression of 3 genes that are related to neovascularization and lymphangiogenesis. Since many genes involve in regulation of neovascularization and lymphangiogenesis, the authors can't make a conclusion that TNFα and its ligands TNF-Rp55 and TNF-Rp75 do not play a significant role. 

Answer: The reviewer is right that we only focused on few markers of neovascularization and lymphangiogenesis. The scope of the study, however, was to assess the differences in standard markers for neovascularization and lymphangiogenesis comparing to different KI models and WT before and after suture placement. We consider VEGF-A/VEGF-C and LYVE-1 as very robust and established markers for these 2 events. A detailed analysis of plenty of different angiogenesis markers like angiopoietin-1 angiopoietin-2, Tie-1, or endothelial cell adhesion molecules like VE-cadherin, PECAM-1, or several integrins would be out of the scope of the manuscript [4]. Furthermore, the take home message of the paper might not benefit from this extension of the analysis due to several problems, e.g. different time-points of regulation and low expression levels that we already suffered from with our rather robust read-out. Taken together, we did not find differential expression of TNFα after suture placement nor any conclusive effects at the level of the TNF-receptors, respectively. 

Q3: TNF-a is an important cytokine in regulating corneal inflammation, neovascularization and lymphangiogenesis. In this study, a suture-induced corneal neovascularization model was used to identify the effect of TNF-a in corneal neovascularization and lymphangiogenesis. The model is so mild, many of the immunological cells and cytokines were not found significantly changed in this model. So that’s possible that you can’t observe much effects of TNF-a in this situation. Also, the suture-induced corneal neovascularization is not suitable to mimic the clinical scenario of corneal graft rejection. As we know, the corneal graft rejection is reasoned by exogenous antigen stimulation, which is not similar to the suture induced corneal neovascularization.

Answer: Indeed the reviewer is right that the suture-induced corneal neovascularization is not the ideal mouse-model to analyze corneal graft rejection, but it mimics the clinical scenario of a high-risk situation for a corneal graft rejection after keratoplasty. Compared to the alkali burn induced corneal neovascularization, the suture-induced model has a number of advantages especially studying the corneal lymphangiogenesis and associated prolymphangiogenic cytokines [5]. First, the sutures are more reproducible than the alkali burns on the cornea. Suturing only causes localized epithelial loss and inflammatory infiltration between the suture and the limbus, but chemical burns deplete the whole epithelial layer of the central cornea and cause strong cellular infiltration of the whole cornea [6]. Therefore, the suture-induced model mimics the clinical scenario of mild corneal lesions better than the alkali burn induced model. Fortunately, chemical burns in patients have become less common.

We clarified this in the discussion part, deleted the chapter about corneal graft rejection and added the following paragraph (p13, lines 363ff).

 “Second, it is more suitable to analyse lymphangiogenesis and associated secretion of prolymphangiogenic cytokines [5]. The induced neovascularisation and lymphangiogenesis mimic the clinical scenario of high-risk situation for a corneal graft rejection after keratoplasty and thus it has more translational impact than the models with severe damage and reduced stimulation of the lymphangiogenesis. Suturing only cause localized epithelial loss and inflammatory infiltration between the suture and the limbus, but chemical burns deplete the whole epithelial layer of the central cornea and cause strong cellular infiltration of the whole cornea [6].” 

Q4: A group of non-injured mice need to be added in each experiment. The authors should identify the expression of TNF-a has been increased in your models. Otherwise, there is no point to investigate the roles of TNF-a and its receptors in this model.

Answer: The reviewer rises two important questions: the lack of an appropriate control and the absence of a significant TNFα increase upon suture placement. 

We have an inert control: only one eye was sutured, the counter-eye served as control. We added a sentence in the method part to highlight the presence of an appropriate control (p6 lines 133f.). The fact that TNFα was not significantly increased in our model does not question our conclusions. First, it might be an issue of the perfect time point that we did not find increased TNFα protein expression. We only checked Day 14 after suture placement. Theoretically, the receptor could also become more sensitive to TNFα at the same concentration by time. We checked not only for TNFα expression, but also TNFα receptor expression as well as function. Thus, we can safely state that TNFα does not play a role in this model. The TNFα increase alone is not necessary or sufficient.

Reviewer #4: The authors show that TNF-α and its receptors are not required for neovascularization and lymphangiogenesis in corneal suture model in mice. The data presented in Fig1 E, F and G indicates that the authors are working with mice where the TNFR receptors are not knocked and thus cannot make any claim on the role of TNFR in angiogenesis and lymphangiogenesis. I have to recommend rejecting the paper in its current form. 

The following issues need to be addressed.

Q1: The quality of tissue in immunofluorescence images Fig1A and S3 and resolution of image is very poor and cannot be interpreted. It is unclear what region of the eye is shown in Fig1A although from result section it appears to be cornea. There is no indication of where TNFα receptor are localizing in the cornea. Please indicate where the receptors are localizing.

Answer: We agree with the reviewer that the resolution of the images was very bad in the draft of our manuscript. We addressed this issue and we now provide significantly better images in S1 Fig (S3) and S2 Fig (S4). We added arrows to depict structures/ staining and we added several remarks in the figure legends to improve understanding. 

Q2: Fig 1B suffers from the same issue as Fig1 A. The tissue section is of extremely poor quality. The limbus region is suspect. Please show a phase image so one might see the iridiocorneal angle. The labels in B and C are confusing is label for C on left makes no sense to the reader. Should it not be cornea? If B and C are from the suture experiment (says Mouse + Suture on the right of panels) where are the controls or sham eyes?

Answer: We agree with the reviewer that the images in Fig1B are lacking the iridiocorneal angle. We display a bigger detail of the images now showing the iridiocorneal to ensure the localization of the section (at the limbus corneae). Furthermore, we are very sorry for the confusion induced by the labelling. All images in Fig1B and C are from WT mice that were sutured. Fig1B illustrates expression of F4/80 and TNFRp55 (left) and TNFrp75 (right) in the limbus area whereas Fig1C shows the expression of F4/80 and TNFRp55 (left) and TNFRp75 (right) in the suture area. We improved the labelling and the figure legends to ease the understanding for the reader here.

Q3: Why is TNFR2 75KDa not reduced in the TNFR2 knockout mouse in Fig 1D? Please reword Line 223-227 “The mRNA expression of TNFRp55 in the TNF-RP55d mice, however, is significantly reduced (Fig.1D, E) and the functionality of the receptors is disturbed. There is no evidence for differential regulation of the other receptor in the TNF-Rpd animals, neither on mRNA nor on protein level (Fig.1D, E,F, G, S1 Table 1) as is very confusing.

Answer: As explained above the mutation in TNFR deficient mice does not lead to a lack of expression, thus, mRNA expression of TNFR2 does not have to be reduced in TNF-Rp75d mice. The explanation is as follows: As already stated above, the TNFR deficiency is a functional one. The mice harbor a null mutation of the respective TNFR leading to an altered gene product that lacks the molecular function of the wild-type gene. The non-functional proteins themselves, however, are expressed. The TNF-Rp55 null mutations were created by disruption of the coding sequence of the TNF-Rp55 gene at base pair 535 (corresponding to the cDNA sequence) by insertion of a neo gene [2]. This resulted in the transcription and transduction of a non-functional TNF-Rp55 protein. Pfeffer et al validated the mutation by an experiment that showed the inability of spleen cells of the TNF-Rp55-/- mice to bind TNFα. Erickson et al, creating the TNF-Rp75-/- mice, pursued a similar strategy [3]. In TNF-Rp75-/- mice challenged by LPS (that leads to shedding of the extracellular domain of TNF-Rp75), no soluble TNF-Rp75 could be detected in the serum. Furthermore, we rephrased the sentence p.9 lines 234-237 accordingly:” The mRNA expression of TNFRp55 in the TNF-RP55d mice, however, is significantly reduced (Fig1D, E). There is no evidence for differential regulation of TNF-Rp75 in TNF-Rp55d animals or vice versa, neither on mRNA nor on protein level (Fig1D, E, F, G, S1 Table)

Q4: Fig 1F and G leads to question the entire premise of the paper- it seems TNFR1 protein is present in mutant hence the authors are working with essentially wt mice?? What is the state of the TNFR2 mutant mice? Does it still express TNFR2 protein? Please validate mice before doing the suture experiments. Please show a western blot showing the protein levels of TNFR1 and TNFR2 in wt and respective mutant mice.

Answer: The reviewer rises several important questions. First, the expression of TNF-Rp55 in TNF-Rp55d mice. As already stated in the answer to Q4 (Reviewer1), the mice harbor null-mutations of the respective genes leading to a malfunction of the receptors and not a lack of expression. Second, the reviewer asks for WB analysis of TNFR1 and TNFR2 expression in the different KI-models/WT mice. Figure 1F depicts an immunoblot showing the protein expression of TNF-Rp55 in TNF-Rp55d mice, TNF-Rp75d mice and wild-type littermates. As shown in bar chart 1G there is no difference in protein expression of TNF-Rp55 among the groups. Unfortunately, we cannot provide a WB of TNF-Rp75 since the antibody did not show a proper signal in WB in several trials.

Q5: Also need to this experiment with the double knockout mice given TNFα knockout seems to exacerbate neovascularization (Fujita 2007).

Answer: Thank you for the recommendation. The TNFα knockout mouse model is a completely different approach compared to our KI models. To evaluate our hypothesis, we however, decided to investigate initially whether one receptor is upregulated when the other is deficient. However, TNF-Rp55 is not upregulated in the TNF-Rp75d mice. Thus, we have convincing evidence, for the lack of compensatory upregulation of the TNFα-receptors expression in the KI-animals on both gene expression and protein expression level. Additionally, TNFα expression remained unchanged after suture placement. Suture placement led to differential regulation of TNFα at the mRNA level, no significant changes were detectable in protein levels, though. No differences were detectable in TNFα comparing WT to mutants. Therefore, we do not expect any influence of TNFα in a double KI or KO model. 

Q6: The figures all seem to be of poor resolution making it hard to zoom in on panels to see detail.

Answer: As already stated above, we substantially improved the resolution of all images and headings.

REFERENCES

1. Khazen W, M'Bika J P, Tomkiewicz C, Benelli C, Chany C, Achour A, et al. Expression of macrophage-selective markers in human and rodent adipocytes. FEBS Lett. 2005;579(25):5631-4. Epub 2005/10/11. doi: 10.1016/j.febslet.2005.09.032. PubMed PMID: 16213494.

2. Pfeffer K, Matsuyama T, Kundig TM, Wakeham A, Kishihara K, Shahinian A, et al. Mice deficient for the 55 kd tumor necrosis factor receptor are resistant to endotoxic shock, yet succumb to L. monocytogenes infection. Cell. 1993;73(3):457-67. Epub 1993/05/07. doi: 10.1016/0092-8674(93)90134-c. PubMed PMID: 8387893.

3. Erickson SL, de Sauvage FJ, Kikly K, Carver-Moore K, Pitts-Meek S, Gillett N, et al. Decreased sensitivity to tumour-necrosis factor but normal T-cell development in TNF receptor-2-deficient mice. Nature. 1994;372(6506):560-3. Epub 1994/12/08. doi: 10.1038/372560a0. PubMed PMID: 7990930.

4. Shih SC, Robinson GS, Perruzzi CA, Calvo A, Desai K, Green JE, et al. Molecular profiling of angiogenesis markers. The American journal of pathology. 2002;161(1):35-41. Epub 2002/07/11. doi: 10.1016/S0002-9440(10)64154-5. PubMed PMID: 12107087; PubMed Central PMCID: PMCPMC1850687.

5. Giacomini C, Ferrari G, Bignami F, Rama P. Alkali burn versus suture-induced corneal neovascularization in C57BL/6 mice: an overview of two common animal models of corneal neovascularization. Exp Eye Res. 2014;121:1-4. Epub 2014/02/25. doi: 10.1016/j.exer.2014.02.005. PubMed PMID: 24560796.

6. Jia C, Zhu W, Ren S, Xi H, Li S, Wang Y. Comparison of genome-wide gene expression in suture- and alkali burn-induced murine corneal neovascularization. Molecular vision. 2011;17:2386-99. Epub 2011/09/17. PubMed PMID: 21921991; PubMed Central PMCID: PMCPMC3171500.

---

## [Decision Letter · Decision Letter 1]

23 Oct 2020

PONE-D-20-16932R1

Effects of TNFα receptor TNF-Rp55- or TNF-Rp75- deficiency on corneal neovascularization and lymphangiogenesis in the mouse

PLOS ONE

Dear Dr. Maier

Thank you for submitting your manuscript to PLOS ONE. After careful consideration, we feel that it has merit but does not fully meet PLOS ONE’s publication criteria as it currently stands. Therefore, we invite you to submit a revised version of the manuscript that addresses the points raised during the review process.

Please pay particular attention to the quality of images and results presented as requested by reviewers. Additionally, discrepancies with published literature should be discussed in the manuscript as requested by reviewers.

We look forward to receiving your revised manuscript.

Kind regards,

Christina L Addison, Ph.D.

Academic Editor

PLOS ONE

Reviewers' comments:

Reviewer's Responses to Questions

**Comments to the Author**

1. If the authors have adequately addressed your comments raised in a previous round of review and you feel that this manuscript is now acceptable for publication, you may indicate that here to bypass the “Comments to the Author” section, enter your conflict of interest statement in the “Confidential to Editor” section, and submit your "Accept" recommendation.

Reviewer #1: (No Response)

Reviewer #2: (No Response)

Reviewer #3: All comments have been addressed

Reviewer #4: (No Response)

2. Is the manuscript technically sound, and do the data support the conclusions?

Reviewer #1: No

Reviewer #2: Yes

Reviewer #3: Yes

Reviewer #4: Partly

3. Has the statistical analysis been performed appropriately and rigorously? 

Reviewer #1: Yes

Reviewer #2: Yes

Reviewer #3: (No Response)

Reviewer #4: Yes

4. Have the authors made all data underlying the findings in their manuscript fully available?

Reviewer #1: Yes

Reviewer #2: Yes

Reviewer #3: (No Response)

Reviewer #4: Yes

5. Is the manuscript presented in an intelligible fashion and written in standard English?

Reviewer #1: Yes

Reviewer #2: Yes

Reviewer #3: (No Response)

Reviewer #4: Yes

6. Review Comments to the Author

Reviewer #1: The authors have improved the manuscript and addressed some of my concerns. However, there are fundamental discrepancies with previous literature that need to be explained prior to publication in PLOS ONE.

B6.129-Tnfrsf1atm1Mak/J, Jackson lab stock 002818 and B6.129S2-Tnfrsf1btm1Mwm/J, Jackson lab stock 002620 are targeted knockout animal strains and do not express TNFRp55 or TNFRp75 respectively.

These are conventional/“traditional” knockout mouse models that were generated in the Mid 90’s using a neomycin resistance gene flanked by homology arms to disrupt gene expression. Although the authors are correct that, in these studies, specific functional assays were used to confirm that the respective genes were no longer functional, this approach leads to a knockout of the targeted gene and not to a mutant form of the targeted gene.

In particular, while the authors indicate that the targeted approach by Pfeffer et al. leads to “the transcription and transduction of a non-functional TNF-Rp55 protein”, Pfeffer et al. generated mice that lack expression of TNFRp55. Similarly, Erickson et al. created mice that do not express TNFRp75. These 2 mouse lines are well-established null/knockout/deficient mouse models that have been utilized extensively. In addition, the knockout strategy is well-described in these two published studies.

With this in mind, the following questions still need to be addressed in a revised manuscript.

Q2: The data in figure 1 would be more convincing if a negative control was included. Since

the authors have access to the TNFR-deficient mice, these samples should be included to

demonstrate specificity of their antibodies.

Q4: It is not clear why the TNFR-deficient mice would show TNFR gene expression. The

authors should validate their RT primers and repeat these experiments.

Q5: Also, the western blot in figure 1F is inconclusive. Why is there no difference between wild type and knock out? Immunoblots for TNFR1 should also be included.

Reviewer #2: Not sure why the labels on the images indicate the fluorphore as opposed to the antigen targeting - please correct.

Adding the negative control to Figure 1, instead of only in a supplement, would still be advisable, but not absolutely necessary.

Reviewer #3: (No Response)

Reviewer #4: The authors have tried to address all the comments to the best of their abilities. However I still have several concerns:

1. My primary issue is the quality of the immunofluorescence data. The quality of the mouse eye sections (they look battered) are still quite terrible and in good conscience cannot allow this to be published . The VEGFA stain is quite peculiar. A schematic to show the location of the suture would be helpful with the limbus and parts of the eye clearly labeled so one may interpret authors images.

2. Please show data points for all graphs.

3. The images on the pdf are still terrible. The quality of the fig page improves when dowloaded. It is quite onerous for the reviewer to download each image (especially under the current pandemic situation and working from home). I don't know why the supplementary images are not part of the document. Please do so in the next iteration.

7. PLOS authors have the option to publish the peer review history of their article (what does this mean?). If published, this will include your full peer review and any attached files.

Reviewer #1: No

Reviewer #2: No

Reviewer #3: No

Reviewer #4: No

---

## [Author Response · Author response to Decision Letter 1]

20 Dec 2020

We thank the editorial board to give us the opportunity to improve our paper substantially according to the reviewers’ comments. 

We answer the reviewer’s concerns point by point as follows.

Reviewer #1: The authors have improved the manuscript and addressed some of my concerns. However, there are fundamental discrepancies with previous literature that need to be explained prior to publication in PLOS ONE.

Q1: B6.129-Tnfrsf1atm1Mak/J, Jackson lab stock 002818 and B6.129S2-Tnfrsf1btm1Mwm/J, Jackson lab stock 002620 are targeted knockout animal strains and do not express TNFRp55 or TNFRp75 respectively.

These are conventional/“traditional” knockout mouse models that were generated in the Mid 90’s using a neomycin resistance gene flanked by homology arms to disrupt gene expression. Although the authors are correct that, in these studies, specific functional assays were used to confirm that the respective genes were no longer functional, this approach leads to a knockout of the targeted gene and not to a mutant form of the targeted gene.

In particular, while the authors indicate that the targeted approach by Pfeffer et al. leads to “the transcription and transduction of a non-functional TNF-Rp55 protein”, Pfeffer et al. generated mice that lack expression of TNFRp55. Similarly, Erickson et al. created mice that do not express TNFRp75. These 2 mouse lines are well-established null/knockout/deficient mouse models that have been utilized extensively. In addition, the knockout strategy is well-described in these two published studies.

With this in mind, the following questions still need to be addressed in a revised manuscript.

We thank the reviewer for his thoroughness concerning the knock-in models. We still think there is a massive misunderstanding. We want to emphasize that B6.129-Tnfrsf1atm1Mak/J, Jackson lab stock 002818 and B6.129S2-Tnfrsf1btm1Mwm/J, Jackson lab stock 002620 lead to the transcription /translation of a truncated protein without proper function and are not KO-mutations in the sense that no mRNA and no protein at all are produced.

It is well known that the classic knockout models produced in the 90s are sometimes not full knockouts in a sense that the mRNA is truncated behind the neomycin cassette and the protein’s length corresponds with nucleotide sequence before the neomycin cassette. In some cases, mRNAS from more distal parts of the gene are produced and lead to a protein that represents a C-terminal part of the protein. Normally in the papers that introduce the model demonstrate the knockout most of the time by mRNA techniques for nucleotide sequences close to neomycin cassette. Thus, we checked this possibility in our study.

As the referee brought up this point in the sense to use TNFRp55d mice as antibody control for the anti-TNFR1 antibody we focus here in our response onto this mouse strain. The evidence comes from our PCR experiments where we show TNFRp55 expression in TNFRp55d mice, when using a primer that binds to nucleotide position 934-957 (F)/1085-1065 (R).

According to the information from Jackson about this strain, the neo cassette is inserted at nucleotide position 535. Using a primer pair that binds to nucleotide 500-520 (F) and 714-733 R or 481-501 (F) and 695-715 R, respectively did not lead to a specific amplification product in TNFRp55d mice, whereas in WT and TNFRp75d it did.

Figure1: mRNA expression of TNFRp55 in WT, TNFrp55d and TNFRp75d. Amplification product size: 234bp (left), 235bp (middle). GAPDH (190bp, right) Served as housekeeping gene. WT P17 O2 retina served as a murine control.

Along with this the immunogen for generating the antibody, we used for IHC and WB (CAT#: AP06465PU-N), Origene is a synthetic peptide, corresponding to the amino acids 370-420 of the human TNF-R1, and corresponding to the C-terminal end of the antibody. Therefore, we cannot use the TNFRp55d mice as an antibody control. 

Thus, we conclude, that it is very likely that TNFRp55 is expressed as a truncated and non-functional protein in B6.129-Tnfrsf1atm1Mak/J.

We base this conclusion both on our experimental evidence and the data from Pfeffer and Erickson.

Our results presented in the manuscript are further substantiated by the new evidence derived from specific mRNA expression in TNFRp55d using primers beyond the neo-cassette and the use of C-terminal amino acids as immunogen.

All the following questions (Q2-Q5) address the same issue: the expression of TNFRp55 in TNFRp55d mice. Thus, for answering the questions we refer to the answer to Q1.

Q2: The data in figure 1 would be more convincing if a negative control was included. Since

the authors have access to the TNFR-deficient mice, these samples should be included to

demonstrate specificity of their antibodies.

Not feasible due to truncated protein and antibody binding at the C-terminal end.

Q4: It is not clear why the TNFR-deficient mice would show TNFR gene expression. The

authors should validate their RT primers and repeat these experiments.

TNF primers validated see above.

Q5: Also, the western blot in figure 1F is inconclusive. Why is there no difference between wild type and knock out? Immunoblots for TNFR1 should also be included.

See above; immunoblot for TNFR1. The western blot shows therefore the truncated protein in the TNFR1-KO, the full protein in the TNFR2 und WT probe. It shows that the TNFR2 knockout does not lead to a compensatory upregulation of the TNFR1.

Reviewer #2: Not sure why the labels on the images indicate the fluorophore as opposed to the antigen targeting - please correct.

Adding the negative control to Figure 1, instead of only in a supplement, would still be advisable, but not absolutely necessary.

Answer: We thank the reviewer for his suggestion. However, we think this is a misunderstanding. Negative control, in this case, means that we incubated the section only with the secondary antibody, omitting the primary antibody, in order to show false positive staining of the secondary antibody. This is the reason why we indicated the fluorophore and not the antigen.

As can be seen in the answer to Reviewer 1, it is impossible to have a TNFR1 negative control using the TNFRp55d mice, since they still express a truncated and non-functional TNFR1 protein, that can be detected by the antibody that binds to an amino acid sequence at the C-terminus. 

Reviewer #3: (No Response)

Reviewer #4: The authors have tried to address all the comments to the best of their abilities. However, I still have several concerns:

Q1. My primary issue is the quality of the immunofluorescence data. The quality of the mouse eye sections (they look battered) are still quite terrible and in good conscience cannot allow this to be published. The VEGFA stain is quite peculiar. A schematic to show the location of the suture would be helpful with the limbus and parts of the eye clearly labeled so one may interpret authors images.

Answer: The quality of the mouse section depends on the processing of the sections. Embedding in paraffin, sectioning, rehydrating and especially the heat mediated antigen retrieval can lead to separations between the different corneal layers, which are very thin in the mouse cornea. Nonetheless, the various corneal layers, epithelium, stroma and endothelium, can be differentiated and are displayed very well in our figures. Indeed, the labeling is not sufficient; therefore we named the anatomic structures and marked the position of the sutures in our Figure 1, Supp. Figure 1 and 4. The limbus region is of particular interest due to the outgrowth of blood and lymph vessels into the cornea to the sutures. 

Q2. Please show data points for all graphs.

Answer: We agree with the reviewer about the importance of showing all data that is included in the experiments to ensure and improve scientific transparency and reproducibility. In our opinion, however, including all data points in the individual graphs would lead to graphs that are very confusing and difficult to interpret. All data points and all raw data analyzed for this manuscript are available at the open access database Zenodo using the following link (https://doi.org/10.5281/zenodo.4049178). If Reviewer 3 is still convinced that the manuscript might profit from adding all data points, we can change the graphs accordingly.

Q3. The images on the pdf are still terrible. The quality of the fig page improves when downloaded. It is quite onerous for the reviewer to download each image (especially under the current pandemic situation and working from home). I don't know why the supplementary images are not part of the document. Please do so in the next iteration.

Answer: We are very sorry for the bad quality of the pdf generated by the submission website. Unfortunately, we are not able to change this. When you download the images, they are in an adequate resolution and quality.

The journal`s guidelines for submissions said to submit the supplement separately. We followed this suggestion. We add the images to the rebuttal letter to ease the review process.

---

## [Editor Report · Decision Letter 2]

23 Dec 2020

Effects of TNFα receptor TNF-Rp55- or TNF-Rp75- deficiency on corneal neovascularization and lymphangiogenesis in the mouse

PONE-D-20-16932R2

Dear Dr. Maier

Thank you very much for your careful consideration of the reviewer's concerns. We’re pleased to inform you that your manuscript has been judged scientifically suitable for publication and will be formally accepted for publication once it meets all outstanding technical requirements.

I would however advise you to carefully scrutinize image quality in the PLoS One system after uploading as indeed the quality of the labels of certain figures render them illegible, and the fluorescent images of those you attached are of much greater quality than those generated in the PDF of the manuscript following upload to the PLoS One Editorial Management system. As such I would contact the journal directly to ensure the image quality in your final manuscript that is to be uploaded online meets your standards and is not impaired by document transfers or resolution issues. 

Kind regards,

Christina L Addison, Ph.D.

Academic Editor

PLOS ONE
---

## [Editor Report · Acceptance letter]

20 Jan 2021

PONE-D-20-16932R2 

Effects of TNFα receptor TNF-Rp55- or TNF-Rp75- deficiency on corneal neovascularization and lymphangiogenesis in the mouse 

Dear Dr. Maier:

I'm pleased to inform you that your manuscript has been deemed suitable for publication in PLOS ONE. Congratulations! Your manuscript is now with our production department. 

Kind regards, 

on behalf of

Dr. Christina L Addison 

Academic Editor

PLOS ONE